# The clinical features and operation experience of multilobar involved congenital lung malformation: A retrospective cohort study

**Kaisheng Cheng**[1], **Miao Yuan**[1], **Gang Yang**[1], **Taozhen He**[1], **Dengke Luo**[1], **Chenyu Liu**[1], **Zongyu Wang**[1], **Jiayin Yang**[2], **Chang Xu**[1]*

1 Department of Pediatric Surgery, West China Hospital, Sichuan University, Chengdu, China, 2 Department of General Surgery, Liver Transplant Center, West China Hospital, Sichuan University, Chengdu, China

* huaxixuchang@163.com

## Abstract

### Background

Congenital lung malformation (CLM) is usually characterized by single-lobe involvement and multilobe involvement is uncommon. There is a lack of experience in the clinical features and surgical treatment of unilateral multilobar CLM. Therefore, this study aims to summarize the clinical characteristics and evaluate the safety and feasibility of thoracoscopic lung sparing resection in the treatment of unilateral multilobar CLM.

### Methods

A retrospective study was conducted on 34 patients with unilateral multilobar CLM, and 34 unilobar patients were randomly selected as the control group in West China Hospital of Sichuan University from 2014.1 to 2021.1. Clinical, operation and follow-up outcomes were compared between the two groups.

### Results

Twelve (35.2%) patients developed preoperative symptoms in the multilobar group more than 4(11.7%) the unilobar group (p = 0.02), the main preoperative symptom was infection; The multilobar group and unilobar group showed significant differences in mean intraoperative blood loss (13.3 mL vs. 7.5 mL; p = 0.02) and mean surgical time (95.6 min vs. 47.5 min; p = 0.037). The median length of postoperative hospital stay and the median chest tube placement time in the multilobar group were significantly longer (5 d vs.3 d, p = 0.045; 2 d vs.1 d, p = 0.031). There were one (2.9%) patient in unilobar group and 4 (11.7%) patients in multilobar group developed complications postoperatively, which is no significant differences between the two groups(p = 0.16). No complications such as thoracic deformity, mediastinal deviation, scoliosis or recurrence were observed in either group at follow-up.

**Data Availability Statement:** All relevant data are within the paper and its Supporting Information files.

**Funding:** XC. This work was supported by the Sichuan Provincial Department of Science and Technology Project [grant numbers No. 2022NSFSC0354]. https://kjt.sc.gov.cn/. There was no additional external funding received for this study. The funders had no role in study design, data collection and analysis, decision to publish, or preparation of the manuscript.

**Competing interests:** The authors have declared that no competing interests exist.

## Conclusions

Unilateral multilobar CLM cases are more susceptible to develop symptoms than unilobar cases. Thoracoscopic lung sparing resection in the treatment of unilateral multilobar CLM has high safety and can effectively avoid serious complications of unilateral multiple lobectomy or pneumonectomy.

## Introduction

Congenital lung malformation (CLM) is a kind of rare congenital disease with abnormal development of the lung airway, lung parenchyma and vessels [1], and mainly includes: congenital pulmonary airway malformation, bronchopulmonary sequestration, etc. In recent years, with the popularization of prenatal screening and improved prenatal examination technology, the detection rate of CLM has gradually increased [2, 3], which has attracted increasing attention from pediatric clinicians.

CLM is usually single-lobe involvement, and multilobe involvement CLM is more infrequent [4]. At present, there is currently no special study to introduce this kind of CLM case. As a result, we know little about the clinical characteristics and treatment strategies of these CLM cases with multilobar involvement, therefore, it is particularly difficult for us to deal with this type of patient.

Because of the high risk of infection in lesions and potential tumor-related risks [5–7], surgical resection has become the recommended treatment for CLM [8]. However, there are several difficulties in the surgical management of these unilateral multilobe lesion CLM cases. Lobectomy is widely accepted as the standard treatment for CLM [8]. Once these patients received lobectomy as a standard procedure, they would have to suffer the multilobes resection or even unilateral pneumonectomy, which is associated with a high mortality rate [9–12], seriously harms the long-term lung function(decreased VC/restricted respiratory dysfunction) [9] and accordingly induces serious postoperative complications, such as postoperative pulmonary edema, bronchopleural fistula, thoracic deformity, mediastinal deviation, and scoliosis in long term follow-up [11–14]. Surgical removal of CLM lesion, while suffering the risk of multilobar resection or pneumonectomy complications, seems not worthwhile. There is very limited experience in treating patients with multi-lobar involvement of CLM, and the research on the treatment of these patients is also unclear. Therefore, there are significant challenges and obstacles in the treatment of CLM patient with multilobar involvement, there is currently no accepted good solution.

Considering these problems, our center applied the one-stage thoracoscopic lung-sparing resection to treat these patients according to the lesion involved scope, which could preserve much more normal lung tissue and may reduce complications compared to multilobes resection or unilateral pneumonectomy.

At present, there is no report about the clinical characteristics of these rare CLM patients and a lack of surgical treatment experience for this kind of patient. Herein we present a series of thoracoscopic lung-sparing resections for unilateral multilobe-involved CLM children performed in a single center since 2014–2021. The aim of this study is to summarize the clinical characteristics of unilateral multilobe-involved CLM cases and evaluate the safety and feasibility of one-stage thoracoscopic lung-sparing resection in the treatment of these patients, and aim to seek optimal clinical management of multilobe-involved CLM.

## Method

### Patients

A retrospective study was conducted on 34 patients with unilateral multilobe involved CLM. Meanwhile, 34 CLM patients with unilobar involvement were randomly selected as the control group in West China Hospital of Sichuan University from January 2014 to January 2021. This study was approved by the Institutional Review Board of West China Hospital of Sichuan University (No.1284). The data in this study were accessed for research purposes from September 3rd, 2022, and this retrospective study used a database from which the patients' identification information had been removed.

In this study, multilobar CLM cases during 2014.1 to 20121.1 were continuously included: preoperative imaging examination, intraoperative exploration and postoperative pathology confirmed multiple lobe involvement, and cases with lost follow-up and incomplete case data were excluded. As a control group, we used a central computer-generated random number list to randomise unilobar patients, selected the same number of unilobar involved CLM cases (as regular cases) from the same period in this study (unilobar vs multilobar).

All patients received preoperative chest tine layer computed tomography (CT) and they received clinical follow up (a radiological evaluation with a chest CT scan) at least 1 year after the operation (3months, 1year). Baseline data and clinical outcomes collected from charts included age at operation, sex, weight, preoperative symptoms, scope of the lesion, intraoperative blood loss, operation time, conversion to open, length of hospital stay, time of chest tube placement, postoperative complications, pathological diagnosis and radiological results in follow-up.

### Operative approach

After satisfactory anesthesia, patients received one-lung or two lung ventilation with artificial pneumothorax (5–10 mmHg, 1L/min $CO_2$ flow). The surgeon used two instruments through two ports, whereas the camera was controlled by the assistant through the third port placed in the middle.

For patients with unilateral unilobar involvement, lobectomy or segmentectomy is performed according to the extent of lesion involvement [15]; For patients with unilateral multilobar involvement, the specific indications for the selection of surgical methods are as follows: according to the lesion involved in each lobe, and deciding whether to perform lobectomy or lung sparing resection(segmentectomy, lesion resection), 1) if the entire lobe of the lung is involved, lobectomy is performed primarily; 2) if only a few lung segments are involved, anatomical segmentectomy is used to remove the lesion segments; and 3) If there is obvious internal and external boundary of the lesion, anatomical lesion resection is used for resection. All operations were completed in one-stage. The specific procedures of lobectomy, segmentectomy and anatomical lesion resection can be found in a previous study [15–17]. According to the involvement of the lesion, the specific surgical methods often include the following categories:

1. multilobe segmentectomy or lesion resection (The lobe with a larger lesion removed first.)

2. lobectomy combined with segmentectomy; lobectomy combined with lesion resection. (The lobectomy was performed first, followed by a segmentectomy or lesion resection)

### Statistical analysis

Continuous variables are presented as the mean ± standard deviation (SD) or as the median (25th–75th percentile) depending on normality. Categorical variables are presented as

numbers (percentages). All statistical analyses were performed using SPSS 21.0 (IBM SPSS Statistics, Chicago, USA).

## Result

From January 2014 to January 2021, 34 multilobar patients and the same amount of unilobar CLMs were included in this study. Baseline data are summarized in Table 1. The median age of the unilobar group was 7 months (range,3–78) and that of the multilobar group was 11 months(range,5–143). The mean BMI was 13.5 in the unilobar group and 16.1 in the multilobar group. As shown in Table 1, the pathological diagnoses were 28 CPAM, 5 IPS and 1 CLE in the unilobar group and 26 CPAM, 3 CPAM+IPS and 5 CLE in the multilobar group. There were no statistically significant differences in age (P = 0.057), BMI (P = 0.12), sex (P = 0.26), or pathological type between the two groups. There were 29(85.2%) patients prenatally diagnosed in the unilobar group and 26(76.4%) patients in the multilobar group (P = 0.35). Four (11.7%) patients developed preoperative symptoms in the unilobar group, and 12(35.2%) patients developed preoperative symptoms in the multilobar group (p = 0.02). In the unilobar group, the main preoperative symptom was infection (4/4); In the multilobar group, 8(8/12) patients developed infection, 2(2/12) patients developed pneumothorax, 1(1/12) patient had hydrothorax, and 1(1/12) patient had hemoptysis.

The operation outcomes are shown in Table 2. The multilobar and unilobar groups showed significant differences in mean intraoperative blood loss (13.3 mL [range, 7–30 mL] vs. 7.5 mL [range, 4–26 mL]; p = 0.02). The mean surgical time was statistically longer in the multilobar group (95.6 min [range, 68–178 min] vs. 47.5 min [range, 29–92 min]; p = 0.037). The median length of postoperative hospital stay in the multilobar group was 5 days (range, 3–15 days) and that in the unilobar group was 3 days (range, 2–8 days) (p = 0.045). The median chest tube placement time in the multilobar group (2 d) was significantly longer than that in the unilobar group (1 d; p = 0.031). Two patients were transferred to open surgery in the multilobar group and no conversion to open surgery was observed in the unilobar group(p = 0.15). Postoperative complications occurred in 4(11.7%) patients in the multilobar group and 1 (2.9%) patient in the unilobar group. In the multilobar group, two patients developed pneumonia, one patient

**Table 1. Clinical characteristics of unilobar and multilobar CLM.**

|  | unilobar (n = 34) | multilobar (n = 34) | P |
|---|---|---|---|
| Median Age (month) | 7(3–78) | 11(5–143) | 0.057 |
| Mean BMI | 13.5±1.3 | 16.1±1.2 | 0.1 |
| Female | 15 | 10 | 0.2 |
| Male | 19 | 24 |  |
| Pathological Type: |  |  |  |
| CPAM[a] | 28 | 26 | 0.54 |
| PS[b] | 5 | 3(cpam+ps) | 0.45 |
| CLE[c] | 1 | 5 | 0.08 |
| Prenatal diagnosis n(%) | 29(85.2%) | 26(76.4%) | 0.35 |
| Preoperative symptom n(%): | 4(11.7%) | 12(35.2%) | 0.02* |
| Infection | 4 | 8 | 0.2 |
| pneumothorax | 0 | 2 |  |
| hydrothorax | 0 | 1 |  |
| hemoptysis | 0 | 1 |  |

a: CPAM congenital pulmonary malformation;b: PS pulmonary sequestration; c: CLM congenital lung emphysema;

**Table 2. Operation outcomes.**

|  | unilobar (n = 34) | multilobar (n = 34) | P |
|---|---|---|---|
| Mean Intraoperative blood loss (ml) | 7.5±4.2 | 13.3±5.2 | 0.02* |
| Mean surgical time (min) | 47.5±10.5 | 95.6±21.7 | 0.037* |
| Median postoperative hospital stay (day) | 3 | 5 | 0.045* |
| Median chest tube drainage time(day) | 1 | 2 | 0.031 |
| Conversion to thoracotomy | 0 | 2 | 0.15 |
| Postoperative Complications n(%): | 1(2.9%) | 4(11.7%) | 0.16 |
| Bronchopleural fistula | 0 | 0 | |
| Hemorrhage | 0 | 0 | |
| Air leakage | 1 | 1 | |
| Atelectasis | 0 | 1 | |
| Pneumonia | 0 | 2 | |
| Recurrence | 0 | 0 | |
| Death | 0 | 0 | |

developed atelectasis and one patient developed air leakage. One patient in the unilobar group developed air leakage, and all these patients recovered after 1–2 weeks of conservative therapy without the need for reoperation. No complications such as bronchopleural fistula, thoracic deformity, mediastinal deviation, scoliosis, death or recurrence were observed in either group during follow-up.

## Discussion

In this study, we were able to assess a relatively large sample size (n = 34) of patients with multilobar involved congenital lung malformation undergoing thoracoscopic resection and the same sample size of regular cases (unilobar involvement CLM). Between the two groups, there were no statistically significant differences in age, BMI, sex, pathological type or prenatal diagnosis rate.

In our study, 29 (85.2%) patients in the unilobar group were prenatally diagnosed, 26 (76.4%) in the multilobar group, there was no significant difference between the two groups. The rate of antenatal diagnosis in the cases with multilobar involvement was no different from that in conventional cases. Four (11.7%) patients developed preoperative symptoms in the unilobar group and 12(35.2%) patients developed preoperative symptoms in the multilobar group (p = 0.02). Patients with multiple lobes involvement were more likely to develop clinical symptoms before surgery than those with unilobar involvement. Priest's study showed that a high proportion of patients (approximately 85.7%) experienced infection symptoms during the preoperative observation period [18]. In our study, the main preoperative symptom was infection (4/4) in unilobar group; In the multilobar group, 8(8/12) patients developed infection, 2(2/12) patients developed pneumothorax, 1 (1/12) patient had hydrothorax and 1 (1/12) hemoptysis. In both groups, the predominant preoperative symptom was infection. In the CLM case of unilateral multilobar involvement, due to a wide range of lesions, there may be a higher risk of infection and an increased risk of respiratory symptoms. Since most of the cystic structures in the lesion replace and squeeze the normal lung tissue, the normal ventilation function is greatly limited [19, 20]. Thus, these patients often require early surgical intervention [6, 21, 22].

In this study, the multilobar and unilobar groups showed significant differences in mean intraoperative blood loss and the mean surgical time was statistically longer in the multilobar group. Considering the involvement of the disease, to perform complete resection, multiple

lobar resection is required for the multilobar group, which will inevitably result in more bleeding and require a longer surgical time, however, it seems that these differences are acceptable in clinical practice. The median chest tube placement time, and the median length of postoperative hospital stay were longer in the multilobar group. Our center routinely does not place drainage tubes after formal thoracoscopic lobectomy [23]. However, because patients in the multilobar group have undergone multilobar resection, there will be more or larger cross-sectional areas of lung tissue after surgery compared to those in the unilobar group (Fig 1). In order to better drain and monitor postoperative complications, the postoperative drainage tube placement time will be longer, and the postoperative hospitalization time will also be correspondingly longer [24]. Two patients were transferred to open surgery in the multilobar group and no conversion to open surgery in the unilobar group. These two patients chose to convert to thoracotomy due to repeated preoperative lung infections, resulting in severe pleural adhesions and a poor visual field [25, 26]. There were no significant differences in postoperative complications between the 4(11.7%) patients in the multilobar group and 1(2.9%) patient in the unilobar group. In the multilobar group, two patients developed pneumonia, one patient developed atelectasis, and one patient developed air leakage. One patient in the unilobar group developed air leakage, and all these patients recovered after 1–2 weeks of conservative therapy without the need for reoperation. Therefore, the complications of lung sparing surgery performed in CLM patients with multilobar involvement are comparable to those of unilobar patients, and this surgical method has high safety for dealing with multilobar involvement CLM patients. The reasons for these results mainly include the following points: First, all lung sparing resection were anatomical resection which can avoid damage to most important structures and reduce postoperative complications [15]. Second, this type of CLM case involves multiple resection surgeries, which are more complex than single lobe surgery. To ensure the safety of surgery, therefore, the surgeon should proficiently master thoracoscopic lobectomy, anatomical segmentectomy, and other thoracoscopic surgeries. In this study, all surgeons underwent the learning curve of those surgeries [27, 28].

No recurrence or residual lesions occurred in either group during the period of hospitalization or follow-up. To avoid residual lesions, in this study, all patients received high-resolution thin-layer CT before surgery to preliminarily assess the extent of lesion involvement, and further determine the extent of lesion through the external boundary of the lesion during surgery (Fig 2). Meanwhile, to achieve complete resection of the lesion, the specific surgical indications are mainly based on the involvement of the lesion in a single lobe, to decide whether to

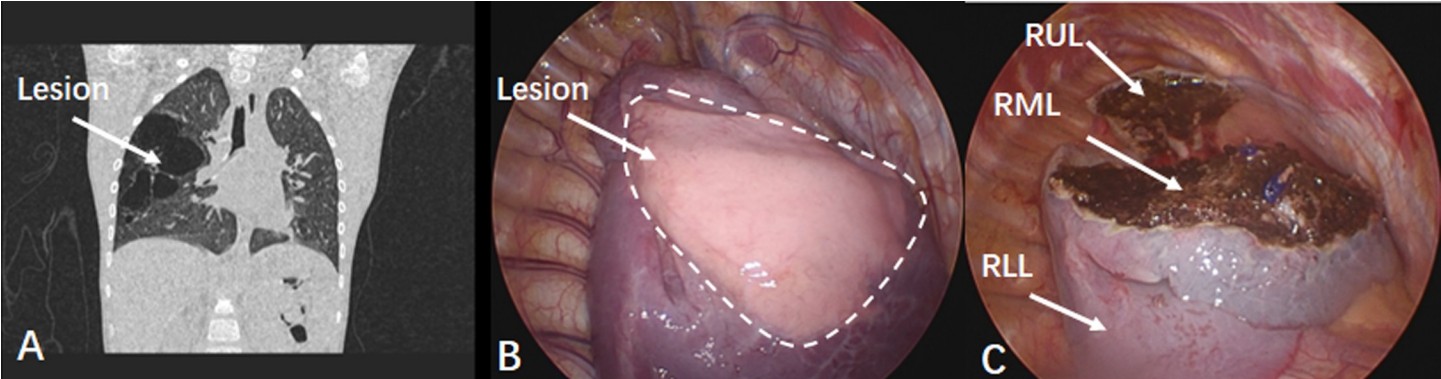

**Fig 1. Preoperative CT scan images and intraoperative images for the patient with right upper and middle lung involvement.** A case with right upper and might lobe involvement received thoracoscopic lung sparing resection. (A)represent the pre-operation CT image of the patient, (B)illustrates the scope of lesion involvement intraoperatively and (C)showed the pulmonary lobe structure in the chest after lesion removal. RUL: right upper lobe, RML: right middle lobe RLL: right lower lobe.

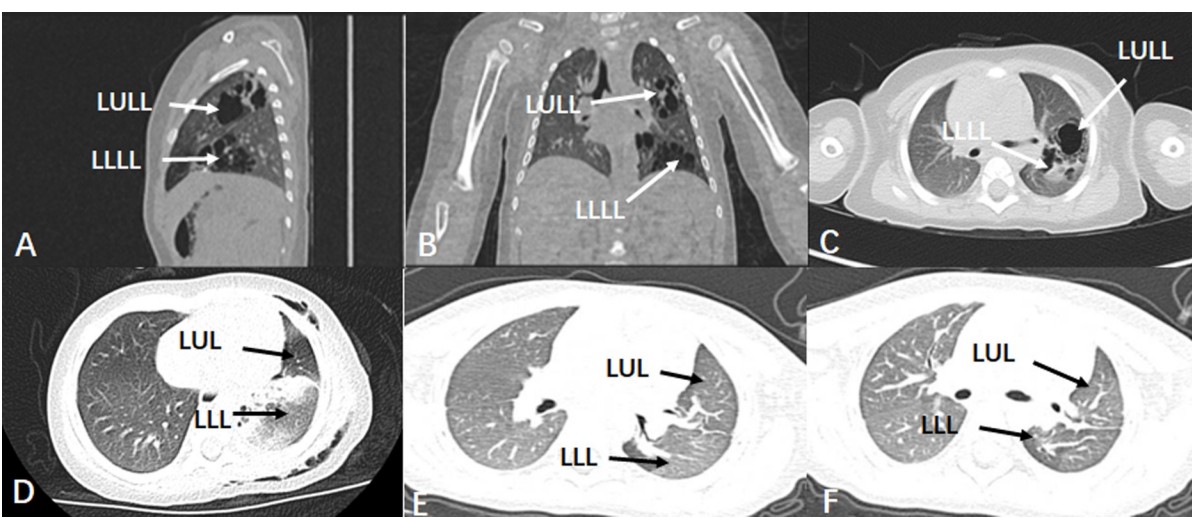

**Fig 2. Preoperative and follow-up CT scan images for the patient with Left upper and lower lung involvement.** A case with upper and lower lobe involvement received thoracoscopic lung sparing resection. (A, B and C) represent the pre-operation CT image of the patient, (D) represent the CT image 1 day after operation, (E) represent the CT image at 3 months after operation and (F) represent the CT images at 1year after operation. LULL: left upper lobe lesion, LLLL: left lower lobe lesion, LUL: left upper lobe, LLL: left lower lobe.

perform lobectomy, segmentectomy, or lesion resection [15]. Besides, in this study, none of the patients experienced long-term complications such as thoracic deformities, mediastinal deviation or scoliosis which may have been caused by unilateral pneumonectomy during a follow-up period of at least 1 year after surgery (Fig 2). Therefore, thoracoscopic lung sparing surgery can effectively remove the lesions while preserving as much normal lung tissue as possible of multilobar involvement CLM patients.

However, our study had several limitations. First, this is a single-center retrospective design study, which may be some selection bias and reduce the generalization of our results. Additionally, because this type of patient is relatively rare in CLM, the sample size was small. Therefore, further studies, including multi-center retrospective or prospective study, are needed. In addition, further research about long-term follow-up, especially pulmonary function, are needed to further determine the effectiveness of the treatment.

## Conclusion

Patients with unilateral multilobar involvement of CLM are more susceptible to develop symptoms than unilobar cases, which may need early intervention. Thoracoscopic lung sparing surgery in the treatment of unilateral multilobar involvement CLM has high safety and may avoid serious complications of unilateral multiple lobectomy or even pneumonectomy, which may be a beneficial surgical treatment.

## Supporting information

**S1 File. The minimal data set of this study.**
(PDF)

## Author Contributions

**Conceptualization:** Kaisheng Cheng, Miao Yuan, Gang Yang, Chang Xu.

**Data curation:** Kaisheng Cheng, Chenyu Liu, Zongyu Wang.

**Formal analysis:** Miao Yuan, Gang Yang, Taozhen He, Dengke Luo.

**Funding acquisition:** Chang Xu.

**Investigation:** Kaisheng Cheng.

**Methodology:** Miao Yuan, Gang Yang, Chenyu Liu, Zongyu Wang.

**Project administration:** Chang Xu.

**Software:** Taozhen He, Dengke Luo.

**Supervision:** Jiayin Yang.

**Visualization:** Jiayin Yang.

**Writing – original draft:** Kaisheng Cheng.

**Writing – review & editing:** Kaisheng Cheng.

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
