## [Decision Letter · Decision Letter 0]

5 Aug 2024

PONE-D-24-25704The clinical features and operation experience ofmultilobar involved congenital lung malformation:A retrospective cohort studyPLOS ONE

Dear Dr. Xu,

Thank you for submitting your manuscript to PLOS ONE. After careful consideration, we feel that it has merit but does not fully meet PLOS ONE’s publication criteria as it currently stands. Therefore, we invite you to submit a revised version of the manuscript that addresses the points raised during the review process.

We look forward to receiving your revised manuscript.

Kind regards,

Luca Bertolaccini, M.D., Ph.D.

Academic Editor

PLOS ONE

Journal Requirements:

XC. This work was supported by the Sichuan Provincial Department of Science and Technology Project [grant numbers No. 2022NSFSC0354].https://kjt.sc.gov.cn/. NO

**Additional Editor Comments:**

The reviewers have emphasized issues that require a careful and thorough revision of the manuscript.

No commitment to publication can be made at this point.

Reviewers' comments:

Reviewer's Responses to Questions

**Comments to the Author**

1. Is the manuscript technically sound, and do the data support the conclusions?

Reviewer #1: Partly

Reviewer #2: Partly

2. Has the statistical analysis been performed appropriately and rigorously? 

Reviewer #1: Yes

Reviewer #2: No

3. Have the authors made all data underlying the findings in their manuscript fully available?

Reviewer #1: Yes

Reviewer #2: Yes

4. Is the manuscript presented in an intelligible fashion and written in standard English?

Reviewer #1: Yes

Reviewer #2: Yes

5. Review Comments to the Author

**Reviewer #1:** Thank you for submitting your manuscript titled “The clinical features and operation experience of multilobar involved congenital lung malformation: A retrospective cohort study” for consideration. I was pleased to receive it as a reviewer.

This is an interesting and potentially valuable study examining the characteristics and surgical outcomes of patients with multilobar congenital lung malformations (CLM) compared to unilobar cases. Strengths of your work include the relatively large sample size for these rare conditions, the detailed comparison of clinical and surgical outcomes between the groups, and the evaluation of lung-sparing surgery for multilobar cases.

To further improve the quality and impact of your manuscript, I have the following suggestions and questions:

1. In the Abstract, consider including the actual percentages or numbers for complication rates, which is a key outcome, rather than just stating there were no significant differences.

2. In the Introduction, could you provide a brief explanation of why multilobar CLM cases are rarer than unilobar cases? This would highlight the importance of your study.

3. Given the rarity of multilobar CLM, have you considered collaborating with other centres to increase your sample size and the generalizability of your findings?

4. Did you follow certain reporting guidelines for your study, such as those provided by the Equator Network?

5. Could you provide more details on how patients were selected for inclusion? Were the patients with multilobar CLM consecutive or was this a convenience sample? What particular method did you use for the random selection of the patients in the unilobar control group? Was this truly random selection or were there any matching criteria used?

6. Consider providing more details on the preoperative workup (e.g., investigations), surgical procedures (e.g., anaesthesia, surgical techniques), and postoperative management of the included patients. Did you follow standardised protocols or were there any differences between groups and individual patients?

7. Could you provide more information on the specific follow-up protocol used? For instance, how frequently were patients seen and what assessments were performed at each visit?

8. In Figure 1, it would be helpful to include labels or arrows pointing out the key features you want readers to notice in the CT images.

9. Have you considered including intraoperative images of interesting multilobar CLM cases? These may have educational value for surgeons.

10. Could you expand on the clinical implications of your findings? How may your results impact decision-making for timing of surgery or choice of surgical approach in multilobar CLM cases?

11. The Discussion could be also strengthened by comparing your results more explicitly with other published studies on surgical management of CLM. How do your outcomes compare to those reported in the literature for traditional lobectomy approaches?

12. You mention in the Discussion that there were “significant differences in mean intraoperative blood loss” between the groups. Although this is true, the clinical significance of 6 ml difference in blood loss is questionable. Could you please acknowledge this or discuss this finding further if you wish?

13. Considering expanding on the limitations of your study (e.g., selection bias due to retrospective design).

14. Consider providing specific directions for future research based on the questions that remain unanswered from your study (e.g., specific CLM, surgical approaches/techniques, outcomes)

15. The conclusion that thoracoscopic lung-sparing surgery “effectively avoids serious complications of unilateral multiple lobectomy or even pneumonectomy” sounds somewhat strong. Could you temper this statement or provide more direct evidence to support it?

16. The manuscript would benefit from a more thorough copy-editing process to address minor grammatical and syntax errors throughout the text.

I believe that addressing these points will further strengthen your contribution to the literature on CLM. I look forward to seeing the revised version of your manuscript.

**Reviewer #2: **Congratulations to the authors for the refined work. The case study is very limited, but certainly also linked to the rarity of the pathology. There should be more emphasis on the description of the complications of CLM if it is not treated surgically and if there is an evolutionary observation study. Review the exposition with more concise sentences and improve the English. Thanks for the contribution

6. PLOS authors have the option to publish the peer review history of their article (what does this mean?). If published, this will include your full peer review and any attached files.

Reviewer #1: **Yes: **Savvas Lampridis

Reviewer #2: No

---

## [Author Response · Author response to Decision Letter 0]

1 Oct 2024

Thank you very much for your kind letter and advice. We have revised the manuscript in accordance with the comments. The detailed responses are as follows:

Reviewer #1:

1.In the Abstract, consider including the actual percentages or numbers for complication rates, which is a key outcome, rather than just stating there were no significant differences.

R: There were one (2.9%) patient in unilobar group and 4(11.7%) patients in multilobar group developed complications postoperatively, which is no significant differences between the two groups(p=0.16). This point has been modified in the manuscript. (Page 2 line 21-24)

2. In the Introduction, could you provide a brief explanation of why multilobar CLM cases are rarer than unilobar cases? This would highlight the importance of your study.

R: At present, there are no studies that explicitly report the specific incidence of multilobar CLM cases. In the past 10 years since 2014, our center has treated more than 1000 patients with CLM, and the cases involving multiple lobes are less than 50 cases, accounting for about 2% of all CLM cases.

3. Given the rarity of multilobar CLM, have you considered collaborating with other centres to increase your sample size and the generalizability of your findings?

R: Thank you for your valuable comments. As you said, this study is only a preliminary summary of our center's experience in the treatment of multi-lobe CLM. Our next research plan will include cases from multiple centers and increase the sample size to make our results more universal.

4. Did you follow certain reporting guidelines for your study, such as those provided by the Equator Network?

R: Yes. This study followed STROBE guidelines for Observational studies.

5. Could you provide more details on how patients were selected for inclusion? Were the patients with multilobar CLM consecutive or was this a convenience sample? What particular method did you use for the random selection of the patients in the unilobar control group? Was this truly random selection or were there any matching criteria used?

R: In this study, the cases in the multi-lobe involvement group were continuous. As a control group, we used a central computer-generated random number list to randomise unilobar patients, selected the same number of unilobar involved CLM cases (as regular cases) from the same period in this study. This point has been modified in the manuscript. (Page 5 line 87-92)

6. Consider providing more details on the preoperative workup (e.g., investigations), surgical procedures (e.g., anaesthesia, surgical techniques), and postoperative management of the included patients. Did you follow standardised protocols or were there any differences between groups and individual patients?

R: We have added the details of anaesthesia, surgical procedures and postoperative management of the included patients. (Page 7 line 104-107) and we believe that patients in each group are following the same protocols and there are no obvious differences between two groups. And the details of the surgical techniques were cited in our center's previous research. [15-17] Considering that the term wedge resection is not quite accurate, we changed it to anatomic lesion resection. （Page8 line 122-126）

7. Could you provide more information on the specific follow-up protocol used? For instance, how frequently were patients seen and what assessments were performed at each visit? 

R: Postoperative patients were generally followed up at 3 months and 1 year. (manuscript page 6 line 97). A chest computed tomography scan was obtained to detect if air leakage or pleural effusion developed, or examine if there were lesion residues, or evaluate the compensation of the remaining lungs. X-ray graph may not effectively and clearly distinguish some subtle signs, so we have used a low-dosed CT examination in the follow up. Three months after surgery, we mainly checked the occurrence of some medium-term complications, and one year after surgery, we mainly concerned the lung compensation and lung function. Meanwhile, we also monitoring some long-term following-up, such as the lung functions of patients (The data of this part is incomplete due to the length of follow-up and will be reported in subsequent studies).

8. In Figure 1, it would be helpful to include labels or arrows pointing out the key features you want readers to notice in the CT images.

R: In order to more vividly express the imaging features of patients with multiple lobes before and after surgery, we added some CT images of patients. Meanwhile. We have added the labels or arrows pointing out the key features. (Page14, 263-265).

9. Have you considered including intraoperative images of interesting multilobar CLM cases? These may have educational value for surgeons.

R: Yes, we have added a figure of intraoperative images of multilobar CLM cases.(Page 13 line 237-243)

10. Could you expand on the clinical implications of your findings? How may your results impact decision-making for timing of surgery or choice of surgical approach in multilobar CLM cases?

R: First of all, in terms of preoperative clinical characteristics of patients with multilobar CLM, it may be significantly less pathogenic than patients with multilobar CLM, and it is more prone to infection than patients with single-lobed CLM during preoperative observation and follow-up. Therefore, from this point of view, early resection is recommended for multilobar CLM patients to avoid complications. Secondly, in terms of surgical methods, lung preservation surgery has its unique advantages for CLM with multi-lobe involvement. It is recommended to perform lung preservation surgery for such patients to better protect the lung function of the remaining lung tissue and increase the long-term lung function of children with CLM. These are shown in the discussion and conclusion sections.

11. The Discussion could be also strengthened by comparing your results more explicitly with other published studies on surgical management of CLM. How do your outcomes compare to those reported in the literature for traditional lobectomy approaches?

R: The purpose of this study was to explore the safety and efficacy of primary lung preservation surgery in patients with multilobar CLM. At present, there are many lung preservation operations for CLM, such as segmental resection, anatomical segmental resection, wedge resection, anatomical lesion resection, etc. The selection of different conservative surgical methods according to different patient conditions is the result of diversification. There may be selection bias in comparing one surgical method alone. Therefore, in this study, the conventional cases with currently recommended lobectomy were used as the control group, aiming to compare the safety and feasibility of lung preservation therapy for special CLM. In the case of multiple lobe involvement, comparison with one surgical method alone may be incomparable and there may be selection bias. Therefore, in this study, the conventional cases with currently recommended lobectomy in our center were used as the control group, aiming to compare the safety and feasibility of lung preservation therapy for special CLM.

12. You mention in the Discussion that there were “significant differences in mean intraoperative blood loss” between the groups. Although this is true, the clinical significance of 6 ml difference in blood loss is questionable. Could you please acknowledge this or discuss this finding further if you wish?

R: In terms of blood loss and even operation time, patients with multiple lobes are indeed more involved than patients with single lobe, but the surgical process of patients with multiple lobes involves the removal of multiple lobular lesions, which is bound to increase in blood loss and operation time compared with cases with single lobe, but for cases with multiple lobes, the average blood loss of 13.3 minutes and the usual operation time of 95.6 minutes are completely acceptable. We have discussed this finding further. (Page 12 line 206-207)

13. Considering expanding on the limitations of your study (e.g., selection bias due to retrospective design).

R: We have expanding the limitation of our study. (Page 14 line267 -270)

14. Consider providing specific directions for future research based on the questions that remain unanswered from your study (e.g., specific CLM, surgical approaches/techniques, outcomes)

R: As multi-lobe CLM case is rare, the treatment strategy for this type of case is still in the exploratory stage. This study preliminarily explored the safety and benefit of lung preservation surgery with a small sample of retrospective study, which seems feasible at present, but the long-term follow-up of patients' lung function and exercise ability is still unclear and in progress. At the same time, it is not clear whether the pathogenesis of this type of patients is the same as that of conventional cases, the incidence of his type of CLM and the best time to operate. (Page15 line271-272)

15. The conclusion that thoracoscopic lung-sparing surgery “effectively avoids serious complications of unilateral multiple lobectomy or even pneumonectomy” sounds somewhat strong. Could you temper this statement or provide more direct evidence to support it?

R: Of course, there is also a chance of serious complications after lobectomy and pneumonectomy, we have our conclusion to be less strong. (Page15 line 277) 

16. The manuscript would benefit from a more thorough copy-editing process to address minor grammatical and syntax errors throughout the text.

R: We have modified the grammatical and syntax errors throughout the text.

Reviewer #2: Congratulations to the authors for the refined work. The case study is very limited, but certainly also linked to the rarity of the pathology. There should be more emphasis on the description of the complications of CLM if it is not treated surgically and if there is an evolutionary observation study. Review the exposition with more concise sentences and improve the English. Thanks for the contribution.

R: Thank you for your recognition of our work, we will continue to strive to improve the management of pediatric CLM. We have modified the grammatical and syntax errors throughout the text.

---

## [Decision Letter · Decision Letter 1]

10 Oct 2024

The clinical features and operation experience ofmultilobar involved congenital lung malformation:A retrospective cohort study

PONE-D-24-25704R1

Dear Dr. Xu,

We’re pleased to inform you that your manuscript has been judged scientifically suitable for publication and will be formally accepted for publication once it meets all outstanding technical requirements.

Kind regards,

Luca Bertolaccini, M.D., Ph.D.

Academic Editor

PLOS ONE

Additional Editor Comments (optional):

Reviewers' comments:

Reviewer's Responses to Questions

**Comments to the Author**

1. If the authors have adequately addressed your comments raised in a previous round of review and you feel that this manuscript is now acceptable for publication, you may indicate that here to bypass the “Comments to the Author” section, enter your conflict of interest statement in the “Confidential to Editor” section, and submit your "Accept" recommendation.

Reviewer #1: All comments have been addressed

Reviewer #2: All comments have been addressed

2. Is the manuscript technically sound, and do the data support the conclusions?

Reviewer #1: Yes

Reviewer #2: Yes

3. Has the statistical analysis been performed appropriately and rigorously? 

Reviewer #1: Yes

Reviewer #2: Yes

4. Have the authors made all data underlying the findings in their manuscript fully available?

Reviewer #1: Yes

Reviewer #2: Yes

5. Is the manuscript presented in an intelligible fashion and written in standard English?

Reviewer #1: Yes

Reviewer #2: Yes

6. Review Comments to the Author

Reviewer #1: Thank you for considering my suggested revisions for your manuscript titled “The clinical features and operation experience of multilobar involved congenital lung malformation: A retrospective cohort study”. It is evident that a significant amount of effort has been directed towards the refining of your work, integrating the feedback provided during the first round of the peer review process. I believe that the resulting changes have significantly improved the rigor and overall quality of your manuscript. I look forward to witnessing the impact your research will have on the clinical and academic community.

Reviewer #2: The work done both for the research and the data exposed respects confidentiality and ethics for publication. Adequate exposition and grammar and good statistical work performed. Starting point and study for further studies and collaborations

7. PLOS authors have the option to publish the peer review history of their article (what does this mean?). If published, this will include your full peer review and any attached files.

Reviewer #1: **Yes: **Savvas Lampridis

Reviewer #2: No

---

## [Editor Report · Acceptance letter]

14 Oct 2024

PONE-D-24-25704R1 

PLOS ONE

Dear Dr. Xu, 

I'm pleased to inform you that your manuscript has been deemed suitable for publication in PLOS ONE. Congratulations! Your manuscript is now being handed over to our production team.

Kind regards, 

on behalf of

Dr. Luca Bertolaccini 

Academic Editor

PLOS ONE